# Novel Therapeutic Strategies for Refractory Ovarian Cancers: Clear Cell and Mucinous Carcinomas

**DOI:** 10.3390/cancers13236120

**Published:** 2021-12-04

**Authors:** Tadahiro Shoji, Shunsuke Tatsuki, Marina Abe, Hidetoshi Tomabechi, Eriko Takatori, Yoshitaka Kaido, Takayuki Nagasawa, Masahiro Kagabu, Tsukasa Baba, Hiroaki Itamochi

**Affiliations:** 1Department of Obstetrics and Gynecology, Iwate Medical University School of Medicine, Iwate 028-3695, Japan; 412.sailing@gmail.com (S.T.); nmrhappy@gmail.com (M.A.); bechitomabehi@gmail.com (H.T.); takatori@iwate-med.ac.jp (E.T.); kaido0428@yahoo.co.jp (Y.K.); shirokuma723@ybb.ne.jp (T.N.); m.kagabu@nifty.com (M.K.); babatsu@iwate-med.ac.jp (T.B.); 2Department of Clinical Oncology, Iwate Medical University School of Medicine, Iwate 028-3695, Japan; itamochi@iwate-med.ac.jp

**Keywords:** ovarian cancer, clear cell carcinoma, mucinous carcinoma, chemotherapy, clinical trial

## Abstract

**Simple Summary:**

Ovarian clear cell and mucinous carcinomas are less sensitive to chemotherapy. This can be explained by carcinogenic mechanisms and molecular biological features. Although chemotherapy with cytotoxic anticancer drugs has been evaluated by clinical studies, none have achieved better treatment outcomes than paclitaxel + carboplatin therapy. In recent years, attention has been focused on treatment with molecular target drugs and immune checkpoint inhibitors that target newly identified biomarkers, and many clinical studies on such treatments have been planned.

**Abstract:**

Ovarian cancer has the worst prognosis among gynecological cancers. In particular, clear cell and mucinous carcinomas are less sensitive to chemotherapy. The establishment of new therapies is necessary to improve the treatment outcomes for these carcinomas. In previous clinical studies, chemotherapy with cytotoxic anticancer drugs has failed to demonstrate better treatment outcomes than paclitaxel + carboplatin therapy. In recent years, attention has been focused on treatment with molecular target drugs and immune checkpoint inhibitors that target newly identified biomarkers. The issues that need to be addressed include the most appropriate combination of therapies, identifying patients who may benefit from each therapy, and how results should be incorporated into the standard of care for ovarian clear cell and mucinous carcinomas. In this article, we have reviewed the most promising therapies for ovarian clear cell and mucinous carcinomas, which are regarded as intractable, with an emphasis on therapies currently being investigated in clinical studies.

## 1. Introduction

The incidence of ovarian cancer is increasing every year. It is one of the most common gynecological malignancies, ranking third after cervical and uterine cancer. In 2017, there were 22,440 estimated new diagnoses of ovarian cancer and 14,080 deaths from the disease in the USA; deaths were higher than those of endometrial and cervical cancer [1].

Epithelial ovarian cancers comprise serous, mucinous, endometrioid, and clear cell carcinomas, and they mimic tissues derived from the Müller duct. In recent years, each of these histological subtypes has been shown to have different carcinogenic mechanisms and molecular biological characteristics. The prognosis of serous ovarian carcinoma has been drastically improved by the identification of biomarkers and development of polyadenosine-diphosphate-ribose polymerase inhibitors [2,3,4]. Compared to the prognosis of serous carcinoma, that of clear cell and mucinous carcinomas is poor [5], a factor attributed to their resistance to chemotherapy.

Highly atypical serous carcinoma exhibits p53 mutations and arises de novo from the superficial epithelium in a short duration. *KRAS* mutations are common in mucinous carcinomas and have been reported to develop according to the adenocarcinoma sequence over a relatively long period of time [6]. Clear cell and endometrioid carcinomas arise from endometriotic ovarian cysts via atypical endometriosis over a long duration [7,8]. Itamochi et al. have demonstrated that the cell doubling time in clear cell carcinomas is approximately twice that of serous carcinomas, suggesting that the low cell proliferative capacity of the clear cell variant is associated with its low sensitivity to chemotherapy [9]. Thus, chemotherapy must be customized to the biological characteristics of each histological type. Since clear cell and mucinous carcinomas are classified as rare tumors, it is difficult to recruit patients for clinical trials and planned studies. Notably, no standard chemotherapy has been established to date.

In this article, we discuss the biological characteristics of ovarian clear cell and mucinous carcinomas and the results of previously reported clinical studies to outline the prospects for new therapeutic strategies.

## 2. Clear Cell Carcinoma

Clear cell carcinoma accounts for approximately 10% of epithelial ovarian cancers in Europe and the United States. However, its incidence in Japan is relatively high, at approximately 25% [10]. The paclitaxel + carboplatin (TC) therapy is the gold standard chemotherapy regimen for ovarian cancers, based on clinical studies including GOG111, OV-10, GOG158, and AGO trials [11,12,13,14]. Paclitaxel stops the process of cell division and kills cancer cells by inhibiting the function of microtubules. On the other hand, carboplatin shows cytotoxicity by inhibiting DNA synthesis, causing cell death in cancer cells [15].

However, in recent years, the sensitivity of clear cell carcinoma to chemotherapy has been reported to be low; different therapeutic strategies have been proposed for each histological subtype [16]. Sugiyama et al. reported that the majority of the patients enrolled in the aforementioned studies had serous carcinoma (from 66% to 72%) and that those with clear cell carcinoma accounted for only 2.1% to 4.9% [17]. Although TC therapy is the standard therapy for poorly and well-differentiated serous carcinomas, including undifferentiated cancers and endometrioid carcinoma, there is no scientific evidence that would necessitate similar treatment for clear cell and mucinous carcinomas. At the fourth Ovarian Consensus Conference held in Vancouver in 2010, an international consensus was reached on the need for separate clinical studies on clear cell carcinoma because each type of ovarian cancer has a different genetic/molecular profile.

### 2.1. Biological Characteristics

The most common genetic mutations in ovarian clear cell carcinoma are adenine thymine-rich interactive domain 1A (*ARID1A*) and phosphatidylinositol 4,5-bisphosphate 3-kinase catalytic subunit alpha (*PIK3CA*) mutations, which have been identified in approximately 50% to 60% of cases. *ARID1A* forms SWI/SNF (switch/sucrose nonfermentable) complexes and disrupts chromatin modeling, consequently causing abnormal expression of various genes. *PIK3CA* contributes to cell survival and proliferation by enhancing the activity of phosphatidylinositol 3-kinase (*PI3K*) to activate the AKT pathway. The mutations in *ARID1A* and *PIK3CA* are frequently known to coexist [18,19]. Moreover, abnormalities in the metabolic pathways are characteristic. Although cancer cells have been confirmed to use anaerobic glycolysis even in an aerobic environment (the Warburg effect), almost all cases of ovarian clear cell carcinoma exhibit high expression levels of hepatocyte nuclear factor (HNF) 1β, which is a main cause of the Warburg effect [20]. Generally, the sensitivity to anticancer drugs is low in hypoxic environments. Abnormalities in the metabolic pathway due to high expression levels of *HNF-1β* may be associated with the resistance of ovarian clear cell carcinomas to anticancer drugs (Table 1).

Tumors with the deficient mismatch repair (dMMR) phenotype respond well to immune checkpoint blockade therapy, as these tumors express many neo-antigens associated with high mutational burden [21]. Therefore, ovarian clear cell carcinomas with *ARID1A* deficiency may benefit from immune checkpoint blockade therapy.

### 2.2. Previous Clinical Studies for Clear Cell Carcinoma

Table 2 summarizes the results of previous clinical studies for ovarian clear cell carcinomas.

Based on preliminary studies conducted in Japan, irinotecan is expected to be effective for clear cell carcinoma. In a preliminary study using γH2AX (a DNA damage marker), Takatori et al. reported that the combination of irinotecan and cisplatin may be effective while assuming that the S-phase arrest and cytotoxic effects of irinotecan are theoretically effective because of the low ratio of S-phase cells and the low growth rate in clear cells [27].

The JGOG3014 trial was a clinical study conducted by the Japanese Gynecologic Oncology Group (JGOG). Targeting patients with ovarian clear cell carcinoma at stages Ic to IV who received initial chemotherapy, this phase II randomized trial compared TC therapy (paclitaxel 180 mg/m^2^ + carboplatin at area under the curve [AUC] 6 on day 1) and irinotecan + cisplatin (CPT-P) therapy (irinotecan 60 mg/m^2^ on days 1, 8, and 15 + cisplatin 60 mg/m^2^ on day 1). CPT-11 is an anticancer drug developed in Japan with the mechanism of action of topoisomerase I inhibition. On the other hand, cisplatin is generally believed to exert its anticancer effects by interacting with DNA, inducing programmed cell death [28,29].

Progression-free survival (PFS) was slightly better with CPT-P therapy, but the difference was not significant. However, the subset analysis showed that the outcomes of CPT-P therapy were better than those of TC therapy in patients with residual tumors less than 2 cm in diameter. This trial also demonstrated the safety of CPT-P therapy, providing the basis for the phase III trial described below [22].

The JGOG3017 trial was a phase III randomized controlled trial for ovarian clear cell carcinomas. Targeting patients at stages I to IV who received postoperative chemotherapy, this trial compared standard TC and CPT-P therapies. In TC therapy, paclitaxel at a dose of 175 mg/m^2^ and carboplatin at AUC 6 were administered on day 1 and repeated every 3 weeks. In the CPT-P therapy, irinotecan (CPT-11) at a dose of 60 mg/m^2^ was administered on days 1, 8, and 15 and cisplatin at a dose of 60 mg/m^2^ was administered on day 1 and repeated every 4 weeks. Both therapies were administered for 6 cycles. In this trial, which enrolled 667 patients, the 2-year disease-free survival rates were 77.6% and 73.0% for TC and CPT-P therapies, respectively (hazard ratio [HR] = 1.17), and the 2-year survival rates were 87.4% and 85.5% (HR = 1.13), respectively. The superiority of CPT-P therapy over TC therapy has not yet been demonstrated [23]. Based on these results, CPT-P therapy is currently used as an alternative to TC therapy in Japan.

The GOG268 trial was a phase II clinical trial in patients with ovarian clear cell carcinoma at stage III/IV who received TC and consolidation therapies combined with temsirolimus. Temsirolimus is a molecularly targeted drug that inhibits cell cycle progression and angiogenesis by blocking mTOR activity [30]. In TC therapy, 1 cycle lasting 3 weeks consisted of the administration of 175 mg/m^2^ paclitaxel and carboplatin at AUC 6 on day 1 in combination with temsirolimus administered at a dose of 25 mg/body on days 1 and 8. Six cycles were performed. Subsequently, consolidation therapy was administered from cycles 7 to 17. During each 3-week cycle of consolidation therapy, temsirolimus was administered at a dose of 25 mg/body on days 1, 8, and 15. This trial enrolled 90 patients, including a total of 45 from the United States and South Korea and 45 from Japan. Exacerbation was detected in 22% of the patients during TC + temsirolimus therapy; 17 cycles of chemotherapy were completed by 28% of the patients. Among the patients with measurable lesions, complete and partial responses were, respectively, observed in 31% (n = 4) and 23% (n = 3) of the patients from the United States and South Korea and 6% (n = 1) and 65% (n = 11) of the patients from Japan. The median PFS and overall survival (OS) were, respectively, 11 and 23 months for the patients from the United States and South Korea and 12 and 26 months for the patients from Japan. In 54% of the patients who underwent optimal surgery, the PFS exceeded 12 months. However, comparison of PFS in the historical control group showed no significant difference [24].

The GOG254 trial examined the effectiveness of sunitinib in 35 patients with recurrence. Sunitinib is a highly potent, selective inhibitor of protein tyrosine kinases, including vascular endothelial growth factor receptor (VEGFR) and platelet derived growth factor receptor (PDGF). In the trial, 1 cycle of sunitinib therapy was set to last 6 weeks–4 weeks of administration at a dose of 50 mg daily, followed by 2 weeks of a resting period. The therapy was administered until disease progression or intolerable toxicity was observed. Although the response rate was 6.7%, 16.7% of patients achieved a PFS of 6 months or longer. The median PFS and OS were 2.7 and 12.8 months, respectively. However, there were patients who developed grade 4 or 5 thrombocytopenia, anemia, acute renal failure, stroke, or allergic reactions [25].

Cabozantinib is an orally bioavailable multitargeted tyrosine kinase inhibitor whose primary targets are MET (IC50 = 1.8 nM), VEGFR2/KDR (IC50 = 0.035 nM) and RET (IC50 = 3.8 nM). The NRG-GY001 trial evaluated the effectiveness of cabozantinib in 13 patients with recurrence. In the trial, 1 cycle of cabozantinib therapy consisted of oral administration of 60 mg once daily for 4 weeks. The therapy was administered until disease progression or intolerable toxicity was observed. A PFS of 6 months or longer was achieved in 23% of the patients, 1 of whom received 23 cycles. The median PFS and OS were 3.6 and 8.1 months, respectively. However, 1 patient developed grade 5 thromboembolism [26].

The results of these trials did not change the standard primary or recurrent treatments for ovarian clear cell carcinoma.

### 2.3. Ongoing or Planned Clinical Studies for Clear Cell Carcinoma

Table 3 summarizes the clinical studies that have completed patient registration and are currently in the analysis stage and those that have been planned to be conducted in the future.

Dasatinib is an oral available short-acting inhibitor of multiple tyrosine kinases. In the GOG283 trial, dasatinib at a dose of 140 mg/body was administered to 35 patients with recurrent clear cell carcinoma of the ovary or endometrium with or without the confirmed expression of BRG-associated factor 250a. One cycle was set to last twenty-eight days, and the drug was administered until disease progression or intolerable toxicity was observed. Case accumulation has been completed, and analyses are currently in progress [31].

Nintedanib is a potent, orally available triple angiokinase inhibitor that targets VEGF, PDGF, and fibroblast growth factor (FGF) receptor signaling pathways [36]. The NiCCC (ENGOT-GYN1) trial is a phase II randomized trial that compares nintedanib with chemotherapy selected by physicians of patients with recurrent clear cell carcinoma of the ovary or endometrium. In the experimental arm, nintedanib at a dose of 200 mg/day will be continuously administered until exacerbation, and the PFS will be compared to that of chemotherapy [32].

Tazemetostat is a first-in-class, selective, oral, mutant and wild-type enhancer of zeste homolog 2 (*EZH2*) inhibitor [37]. In the NRG-GY-014 trial, a 28-day cycle of tazemetostat therapy will be repeated for recurrent clear cell carcinomas of the ovary and endometrium, unless disease progression or intolerable toxicity is observed. The primary endpoint is the response rate, and this trial is planned to include 86 patients [33].

Epacadostat decreases tryptophan metabolism by inhibiting IDO1, which results in the enhanced proliferation of effector T cells and natural killer cells, the decreased apoptosis and increased activation of CD86^high^ dendritic cells, and the reduced expansion of regulatory T cells [38]. The anti-PD-1 antibody pembrolizumab was tested in clinical trials as maintenance therapy for ovarian cancer [39]. In preclinical models, epacadostat plus an ICI suppressed tumor growth more effectively than single-agent treatment, primarily through reactivation of antitumor immunity [40]. In the NRG-GY-016 trial, which targets patients with recurrent clear cell carcinoma, epacadostat is orally administered on consecutive days and pembrolizumab was administered every 21 days. One cycle is set to last twenty-one days and repeated until disease progression or intolerable toxicity is observed. The primary endpoint is the response rate, and the secondary endpoints are safety, PFS, and OS. This trial enrolled 14 patients, and the enrollment was closed in February 2021 [34].

Ceralasertib (AZD6738), a potent, selective ataxia telangiectasia and Rad3-related kinase inhibitors, is under evaluation as monotherapy and in combination with chemotherapy, ionizing radiation, immunotherapy, and other anti-cancer drugs, including PARP inhibitors in a variety of genetic contexts [41]. The ATARI trial (ENGOT/GYN1/NCRI) is a phase II randomized trial comparing response rates between ceralasertib alone and in combination with olaparib in patients with gynecologic cancers, including recurrent clear cell carcinoma, with or without a confirmed *ARID1A* deficiency. This trial is planned to include at least 40 patients [35].

### 2.4. New Pharmacotherapy for Clear Cell Carcinoma

Immune checkpoint inhibitors are promising drugs. In a phase II clinical study of nivolumab (an anti-programmed cell death 1 antibody) involving patients with platinum-resistant recurrent ovarian cancer, Hamanishi et al. reported that 2 of 20 patients achieved complete response. Clear cell carcinoma was diagnosed in one of the two patients, and serous carcinoma was diagnosed in the other. Notably, the latter had tumors with a gene expression profile similar to that of clear cell carcinoma [42]. Since this clinical study included only two patients with clear cell carcinoma, this histological type may be more sensitive to immune checkpoint inhibitors than other types.

Some PI3K pathway inhibitors using *PIK3CA* mutations as a target may be identified as effective drugs. TAS-117, an AKT inhibitor, is currently being investigated in a phase I clinical study on patients with clear cell carcinoma, including those positive for *PIK3CA* mutations [43].

Although the rate of *TP53* mutation is low in clear cell carcinomas, the murine double minute 2 (*MDM2*) gene, which is involved in *TP53* degradation, is more frequently expressed in this histological type than in other types. In clear cell carcinoma lines without *TP53* mutations, *MDM2* inhibitors have exhibited antitumor effects both in vitro and in vivo [44]. Consequently, *MDM2* inhibitors are currently being investigated in clinical studies.

*ARID1A* deficiency is observed in gynecological cancers such as ovarian cancer. *ARID1A* gene mutations can also be identified by gene panel testing. Metabolome-targeted anticancer drugs are still unexplored, but they are promising and new drug discovery, and personalized treatment is expected [18].

## 3. Mucinous Carcinoma

### 3.1. Biological Characteristics

Many cases of ovarian mucinous carcinoma have been reported to metastasize from gastrointestinal carcinomas [45]. In Japan, primary ovarian mucinous carcinoma accounts for 11% of epithelial ovarian cancer cases [46]. In a clinical study conducted in patients with epithelial ovarian cancer at the International Federation of Gynecology and Obstetrics (FIGO) stages III to IV by the Gynecologic Oncology Group (GOG), 54 patients (1.5%) had mucinous carcinoma. Of the 44 patients who underwent central pathological review, approximately 60% had metastatic ovarian cancer [47]. In a multicenter clinical study conducted in Japan, 87 of 151 patients (57.6%) with ovarian mucinous carcinoma had metastatic ovarian cancer [48]. Thus, for the diagnosis of primary ovarian mucinous carcinoma, it is important to exclude metastatic variants and to check the presence or absence of interstitial infiltrate while monitoring the tumor diameter and the characteristics of individual lesions [49].

In ovarian mucinous carcinoma, abnormalities in the tumor protein p53 (*TP53*) gene, which are frequently observed in serous carcinoma, are relatively uncommon [34]. Abnormalities in the *KRAS* gene, which are common in gastrointestinal carcinoma, were observed in 32–56% of patients [50,51,52,53], and the human epithelial growth factor receptor 2 (*HER2*) gene was amplified in 18% [54]. However, no abnormalities were observed in the V-raf murine sarcoma viral oncogene homolog B1 (*BRAF*) gene [53]. Table 4 summarizes the frequencies of genetic abnormalities in mucinous ovarian and colorectal carcinomas [50,51,52,53,54,55,56,57,58,59,60]. Since ovarian mucinous carcinomas frequently express proteins such as cytokeratin (CK) 7, CK 20, caudal-related homeobox transcription factor (CDX) 2, mucin (MUC) 2, and MUC5AC, they are suggested to have biological characteristics similar to those of gastrointestinal carcinomas [61,62]. Table 5 summarizes the frequencies of protein expression in mucinous ovarian and colorectal carcinomas [63,64,65,66,67,68,69,70,71,72,73].

### 3.2. Chemotherapy for Mucinous Carcinoma

Table 6 shows the previously reported outcomes of chemotherapy [48,74,75,76,77]. In a study involving 27 patients with mucinous carcinoma and 54 patients with serous carcinoma, the response rates to chemotherapy, disease-free survival, and OS were all worse in the latter [74]. In a retrospective study comprising 420 patients with epithelial ovarian cancer at FIGO stages III to IV, who received combination chemotherapy with paclitaxel and platinum-containing drugs, 24 patients with mucinous carcinoma exhibited a significantly lower response rate to chemotherapy (45% vs. 87%) and shorter median survival (15.4 months vs. 47.7 months) than 367 patients with serous carcinoma [75]. In a Japanese study conducted by Shimada et al., the response rate to chemotherapy in patients with mucinous carcinoma was 12.5%, which was significantly lower than the 68.4% in patients with serous carcinoma [48]. Thus, TC therapy, which is the standard chemotherapy for epithelial ovarian cancer, cannot be considered an effective treatment option for mucinous carcinoma, thereby necessitating the urgent development of new therapies.

Since mucinous ovarian and gastrointestinal carcinomas have similar biological characteristics, the regimens used for the latter have been attracting attention. Oxaliplatin (L-OHP) is a platinum-based anti-cancer drug and we have reported its efficacy for adenocarcinoma of the uterine cervix. L-OHP binds to the DNA of cancer cells and induces DNA replication and apoptosis of cancer cells, resulting in an anti-tumor effect [78]. 5-Fluorouracil (5-FU) is a fluoropyrimidine antimetabolism agent that exhibits antitumor effects by inhibiting DNA synthesis [79]. Using cell lines derived from ovarian mucinous carcinoma, Sato et al. demonstrated that a combination of L-OHP and 5-FU inhibited cell proliferation by 50% or more in four of five cell lines. They also reported that the combination therapy significantly increased the survival of nude mouse models of cancerous peritonitis, compared with L-OHP or 5-FU monotherapy [80]. Thus, combining L-OHP and 5-FU is a promising strategy for the treatment of ovarian mucinous carcinoma.

### 3.3. Clinical Studies Using L-OHP and 5-FU for Mucinous Carcinoma

Kurnit et al. reported a retrospective cohort study of 52 patients with ovarian mucinous carcinoma that compared 5-FU, capecitabine, and L-OHP in 26 patients treated with the regimens for gastrointestinal carcinoma and 26 patients treated with the regimens for ovarian cancer. In this study, the gastrointestinal carcinoma regimens improved OS in patients with ovarian mucinous carcinoma requiring postoperative chemotherapy (HR: 0.2, 95% confidence interval: 0.1–0.8, *p* = 0.01) [81].

The mEOC/GOG241 trial was a phase III randomized control trial that targeted patients with ovarian mucinous carcinoma at stages II to IV undergoing initial treatment and those with recurrent ovarian mucinous carcinoma at stage I who had no history of chemotherapy. This trial compared TC therapy with L-OHP + capecitabine combination therapy (XELOX therapy). Capecitabine is a prodrug of 5-FU. These therapies were then compared after the addition of the angiogenesis inhibitor bevacizumab. This trial compared and analyzed only 50 patients because of the delayed accumulation of cases. Compared with TC therapy, XELOX therapy did not improve survival (HR = 0.78), even after the addition of bevacizumab (HR = 1.04) [82].

In Japan, a phase II clinical study was conducted on S-1 + L-OHP therapy for advanced and recurrent ovarian mucinous carcinoma. S-1 is also a prodrug of 5-FU. The primary endpoint was the response rate, and the secondary endpoints were the incidence of adverse events, PFS, and OS. The response and disease control (which included stable disease) rates were 12% and 70%, respectively. These results demonstrated the importance of salvage chemotherapy for advanced and recurrent cancers, which are considered to have poor prognoses. However, based on the central pathological review, metastatic cancer was diagnosed in 19 of the 33 patients. This reaffirmed the low rate of accurate diagnosis of primary mucinous carcinoma [83].

However, these anticancer drugs did not change the standard primary or recurrent treatments for ovarian mucinous carcinoma.

### 3.4. New Pharmacotherapy for Mucinous Carcinoma

Molecular target drugs have been attracting attention as new therapies for mucinous carcinomas, which are resistant to platinum-based chemotherapy [49]. Compared to colorectal cancer, *HER2-neu* gene amplification is relatively more common in ovarian mucinous carcinoma, it responds to trastuzumab, a humanized monoclonal antibody targeting *HER2* either alone or in combination with oral lapatinib, a tyrosine kinase inhibitor [54,84]. Furthermore, an in vivo study showed that cetuximab, an epithelial growth factor receptor inhibitor, is effective for the treatment of mucinous carcinomas without *KRAS* mutations [85]. Another study demonstrated that this inhibitor exerts a synergistic effect by inhibiting mitogen-activated protein kinase (*MEK*) and PI3K in ovarian mucinous carcinoma cell lines with *KRAS* mutations [86]. In a phase I clinical study of a rare ovarian cancer subtype, Spreafico reported that favorable objective responses were obtained by simultaneous inhibition of *MEK* and *PI3K* in patients with *KRAS* mutation-associated ovarian cancer [87]. The use of such molecular target drugs is expected to facilitate the development of new therapeutic strategies for advanced and recurrent ovarian mucinous carcinomas.

## 4. Conclusions

Ovarian cancer is diverse at the molecular level, and clear cell and mucinous carcinomas exhibit low sensitivity to chemotherapy. Although chemotherapy regimens for ovarian clear cell and mucinous carcinomas have been evaluated by numerous clinical studies, they have failed to exhibit treatment outcomes superior to those of TC therapy. The identification of biomarkers and development of therapeutic drugs specific to each type of ovarian cancer are anticipated. For ovarian clear cell carcinoma, in which the PI3K/AKT/mTOR pathway and the *MDM2* gene are prognostic factors, AKT and *MDM2* inhibitors may prove to be promising therapeutic drugs in the future. The biomakers for ovarian mucinous carcinoma, including the *KRAS* and *HER2-neu* genes, *MEK*, and *PI3K*, and molecular target drugs such as trastuzumab, lapatinib, and cetuximab have been gaining attention. We hope that molecular target drugs and immune checkpoint inhibitors targeting these genomic alterations will be developed and clinically applied in the future.

## Figures and Tables

**Table 1 cancers-13-06120-t001:** Summary of critical genetic changes in ovarian clear cell carcinoma.

Gene	Pathways Affected	Type of Expression Abnormality	Frequency (%)
ARID1A [18]	SWI/SNF	Mutation	62
ARID1B [18]	SWI/SNF	Mutation	10
PIK3CA [18,19]	PI3K	Mutation	35–51
PTEN [18,19]	PI3K	Mutation	2–5
PIK3R1 [18,19]	PI3K	Mutation	7–8
PIK3R2 [19]	PI3K	Mutation	5
KRAS [19]	MAPK	Mutation	9
ERBB2 [19]	MAPK	Mutation and amplification	11
HNF-1β [20]	Metabolic pathway	Methylation	>80

Abbreviations: ARID1A, AT-rich interactive domain 1A; ARID1B, AT-rich interactive domain 1B; ; PIK3CA, phosphatidylinositol-4,5-bisphosphate 3-kinase catalytic subunit alpha; PTEN, phosphatase and tensin homolog; PIK3R1/2, phosphoinositide-3-kinase regulatory subunit 1/2; ; KRAS, KRAS proto-oncogene; ERBB2, erb-b2 receptor tyrosine kinase 2; HNF1β, hepatocyte nuclear factor 1 homeobox B; SWI/SNF, switch/sucrose non-fermentable; PI3K, phosphatidylinositol 3-kinase; MAPK, Mitogen-Activated Protein Kinase.

**Table 2 cancers-13-06120-t002:** Previous clinical trials for ovarian clear cell carcinoma.

Trials	Patients	N	Arms/Treatments	ORR	Median PFS/ 2-Year Disease-Free Survival Rate ***	Median OS/ 2-Year Survival Rate ***
JGOG3014 [22]	Stage I-IV First-line	99	TC* × 6 CPT-P × 6	40 25	NA NA	NA NA
JGOG3017 [23]	Stage I-IV First-line	667	TC** × 6 CPT-P × 6	46.7 29.4	77.6% *** 73.0% ***	87.4% *** 85.5% ***
GOG268 [24]	StageIII/IV First-line	45 (Japan) 45 (US/Korea)	TC** + Temsirolimus 25 mg/body × 6 →Temsirolimus 25 mg/body	71 (Japan) 54 (US/Korea)	12 (Japan) 11 (US/Korea)	26 (Japan) 23 (US/Korea)
GOG254 [25]	Recurrent	35	Sunitinib 50 mg/day	6.7	2.7	12.8
NRG-GY001 [26]	Recurrent	13	Cabozantinib 60 mg/day	0	3.6	8.1

Abbreviations: TC*, Paclitaxel 180 mg/m^2^, Carboplatin AUC6 on day1 every 3weeks; TC**, Paclitaxel 175 mg/m^2^, Carboplatin AUC6 on day1 every 3 weeks; CPT-P, Irinotecan 60 mg/m^2^ on days1,8,15, Cisplatin 60 mg/m^2^ on day1 every 4 weeks; ORR, Objective response rate; PFS, Progression-free survival; OS, Overall survival. *** *p* < 0.005.

**Table 3 cancers-13-06120-t003:** Ongoing or planned clinical studies for ovarian clear cell carcinoma.

Table.	Patients	Phase	N	Arms/Treatments
GOG283 [31]	Recurrent	II	35	Dasatinib 140 mg/day
NiCCC (ENGOT-GYN1) [32]	Recurrent	II (Randomized)	120	SoC Nintedanib 400 mg/day
NRG-GY-014 [33]	Recurrent	II (basket)	86	Tazemetostat
NRG-GY-016 [34]	Recurrent	II	14	Pembrolizumab + Epacadostat
ATARI [35]	Recurrent	II	40<, <116	Ceralasertib 160 mg +/− Olaparib 600mg/day *

Abbreviations: SoC, Standerd of care[ Ovarian Cancer Patients: Paclitaxel (80 mg/m^2^) IV Day 1, 8, 15 every 28 days Pegylated Liposomal Doxorubicin (PLD) (40 mg/m^2^) IV every 28 days Topotecan (4 mg/m^2^) IV Day 1, 8, 15 every 28 days,Endometrial Cancer Patients: Carboplatin (AUC 5) and Paclitaxel (175 mg/m^2^) IV every 21 days Doxorubicin IV (60 mg/m^2^) every 21 days]; * Cohort 1A patients receive ceralasertib monotherapy (160 mg tablets twice daily on days 1–14 in a 28 day cycle). If no activity is observed in this cohort, cohort 1B will open, with the same patient population receiving ceralasertib plus olaparib in combination (160 mg ceralasertib tablets once daily on days 1–7 and 300 mg olaparib tablets twice daily continuously in a 28 day cycle). Patients with clear cell carcinomas (ovarian, endometrial, or endometriosis related) with no ARID1A loss enter cohort 2 and patients with other relapsed gynecological subtypes enter cohort 3, irrespective of ARID1A status. Both cohort 2 and cohort 3 patients receive combination therapy (160 mg ceralasertib tablets once daily on days 1–7 and 300 mg olaparib tablets twice daily continuously in a 28 day cycle).

**Table 4 cancers-13-06120-t004:** Frequency of molecular alterations in ovarian and colorectal mucinous carcinomas .

	Primary Ovarian Mucinous Carcinomas	Primary Colorectal Mucinous Carcinomas
KRAS mutations	32–56% [50,51,52,53]	23–38% [55,56,57]
HER2 amplification	18% [54]	<1% [58]
BRAF mutations	0% [53]	14–28% [56,57,59]
TP53 mutation	26% [50]	20–48% [59,60]

Abbreviations: KRAS, V-Ki-ras2 Kirsten rat sarcoma viral oncogene homolog; HER2, human epithelial growth factor receptor 2; BRAF, V-raf murine sarcoma viral oncogene homolog B1; TP53, Tumor protein p53.

**Table 5 cancers-13-06120-t005:** Frequency of expression of selected markers used for differential diagnosis of ovarian and colorectal mucinous carcinomas.

.	Primary Ovarian Mucinous Carcinomas	Primary Colorectal Mucinous Carcinomas
CK7	79–100% [63,64,65]	10% [69]
CK20	56–98% [64.65]	100% [64]
CDX2	18–42% [63,66]	59% [70]
MUC2	100% [67]	86–96% [67,71,72,73]
MUC5AC	50–100% [64,68]	2–33% [70,73]

Abbreviations: CK7, Cytokeratin 7; CK20, Cytokeratin 20; CDX2, Caudal-related homeobox transcription factor 2; MUC2, Mucin 2; MUC5AC, Mucin 5AC.

**Table 6 cancers-13-06120-t006:** Previous reports of chemotherapy for mucinous carcinoma.

Author	Patients	N	Regimen	OOR (%)	Median PFS	Median OS
Shimada M [48]	Stage I-IV	24	Platinum based regimen	12.5	NA	NA
Hess V [74]	Stage III/IV	19	Platinum based regimen	26.3	5.7	12.0
Bamias A [75]	Stage III/IV	24	Paclitaxel/platinum	45.0	NA	15.4
Pectasides D [76]	Stage III/IV	47	Platinum based regimen	38.5	11.8 (TTP)	33.2
Pisano C [77]	Stage I-IV	19	Platinum based regimen	42.1	NA	NA

Abbreviations: ORR, Objective response rate; PFS, Progression-free survival; OS, Overall survival; NA, not available; TTP, Time to progression.

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
