# Peer review of "Novel Therapeutic Strategies for Refractory Ovarian Cancers: Clear Cell and Mucinous Carcinomas"

_cancers, 2021, doi:10.3390/cancers13236120_

Round 1

Reviewer 1 Report

This manuscript describes the status of current clinical trials and potential targeted and immunotherapies for clear cell and mucinous ovarian cancer, two subtypes that do not respond well to standard of care chemotherapeutic regimens. This review is generally well structured and contains informative tables. Overall, I recommend that the authors expand on some of the biological mechanisms of resistance to conventional chemotherapies. Furthermore, a more comprehensive discussion of relevant in vitro and in vivo models (e.g. cells lines, organoids, xenografts, etc.) to validate potential targets and provide rational therapeutic strategies would strengthen this review. I have some additional comments.

  1. Lines 86-92. It’s unclear as to why the authors decide to focus on HNF. Some context and expansion of discussion for its importance would be helpful.
  2. Lines 94-99. The description of irinotecan here seems out of place and no context is provided. This could be moved to section 2.2.
  3. In Section 2.4, please include a discussion of potential targets for ARID1A (e.g. epigenetic inhibitors).
  4. Line 209. Are there any MDM2 in clinical trials? If so, these should be included into Table 3.
  5. Lines 256-261. Please elaborate on how why L-OHP and 5-FU is more effective than standard chemotherapy.

Author Response

Response to Reviewer1

"This manuscript describes the status of current clinical trials and
potential targeted and immunotherapies for clear cell and mucinous
ovarian cancer, two subtypes that do not respond well to standard of
care chemotherapeutic regimens. This review is generally well structured
and contains informative tables. Overall, I recommend that the authors
expand on some of the biological mechanisms of resistance to
conventional chemotherapies. Furthermore, a more comprehensive
discussion of relevant in vitro and in vivo models (e.g. cells lines,
organoids, xenografts, etc.) to validate potential targets and provide
rational therapeutic strategies would strengthen this review. I have
some additional comments.

Thank you for reviewing our paper and for your appropriate comments. We responded to each comment. The revised parts of the text were also indicated in red.

    Lines 86-92. It’s unclear as to why the authors decide to focus on HNF. Some context and expansion of discussion for its importance would be helpful.

Thank you for your comments. Overexpression of HNF-1β may play a role in the mechanism of anti-cancer drug resistance, which is important in considering the biological characteristics of ovarian clear cell carcinoma. Therefore, we have listed HNF-1β in the text and Table 1..

    Lines 94-99. The description of irinotecan here seems out of place and no context is provided. This could be moved to section 2.2.

Thanks for the advice. This sentence has been moved to Section 2.2.

    In Section 2.4, please include a discussion of potential targets for
ARID1A (e.g. epigenetic inhibitors).

Thank you for your comments. We have added a discussion on ARID1A to Section 2.1 and 2.4.

    Line 209. Are there any MDM2 in clinical trials? If so, these should
be included into Table 3.

Unfortunately, we were unable to find any ongoing clinical trials using MDM2 inhibitors for ovarian clear cell carcinoma.

    Lines 256-261. Please elaborate on how why L-OHP and 5-FU is more
effective than standard chemotherapy."

Thanks for the comments. The molecular biology of ovarian mucinous carcinoma is similar to that of gastrointestinal carcinoma. Therefore, L-OHP and 5-FU, which are effective in gastrointestinal cancers, are thought to be effective, and we conducted a phase II clinical trial of SOX therapy. However, there is no clear evidence of molecular biological mechanism.

Reviewer 2 Report

Shoji et al. described the recent clinical trials for clear cell and mucinous carcinomas in this manuscript. The molecular targets of these two cancer types were also listed. They also pointed out the molecular target drugs and immune checkpoint inhibitors targeting prognostic markers are promising for the future treatments of these cancers.

The following points should be addressed in the manuscript:

  1. How do autologous stem cell transfer and immune therapy work in these two types of ovarian cancers? Both have been used for other ovarian cancers in clinic.
  2. Authors should explain and describe the mechanisms of action of every drug mentioned in the paper including paclitaxel, carboplatin, irinotecan… whether they are new versions of an older drug or are novel compounds, and what their molecular targets are. Some of the drugs were described, but many were not in the whole manuscript.
  3. There should be a brief summary of the results in the end of each section. It is hard to follow or get an idea of which trials failed completely and the reasons, which one advanced to the clinic or further (combination) trails (include the trail numbers), and what was the most promising treatment. For example, authors can add another column in the tables showing the next step, either proceed to clinic or ended without proceeding for each trial.
  4. Authors should include rates of submission and success of clinical trials: how many percent of the drugs entering clinical trials that make it to clinical use? How many percent of the clinical trials for ovarian cancers is targeting clear cell and mucinous carcinomas?
  5. Authors mentioned the deficiency of primary diagnosis which can lead to the failure of treatments. They should propose new strategies for increasing accurate primary diagnosis in the conclusions.

Author Response

Response to Reviewer2

Shoji et al. described the recent clinical trials for clear cell and mucinous carcinomas in this manuscript. The molecular targets of these two cancer types were also listed. They also pointed out the molecular target drugs and immune checkpoint inhibitors targeting prognostic markers are promising for the future treatments of these cancers.

The following points should be addressed in the manuscript:

Thank you for reviewing our paper and for your appropriate comments. We responded to each comment. The revised parts of the text were also indicated in red.

1,How do autologous stem cell transfer and immune therapy work in these two types of ovarian cancers? Both have been used for other ovarian cancers in clinic.

Thank you for your comments.

Immuno therapy is effective for ovarian clear cell carcinoma. Tumors with the dMMR phenotype respond well to immune checkpoint blockade therapy, as these tumors express many neo-antigens associated with high mutational burden [70]. BAF250A/ARID1A protein interacts with the MMR protein MSH2 and promotes MMR. Therefore, ARID1A deficiency might be an indicator of the dMMR phenotype, which is linked to the efficacy of immune checkpoint blockade therapy. (Ref21)

We added reasons why ovarian clear cell carcinomas are suitable for immune therapy in the sub-section2.1. We added these to Section 2.1.

2,Authors should explain and describe the mechanisms of action of every drug mentioned in the paper including paclitaxel, carboplatin, irinotecan… whether they are new versions of an older drug or are novel compounds, and what their molecular targets are. Some of the drugs were described, but many were not in the whole manuscript.

Thank you for your advice. We have added the mechanism of action of every drug.

3,There should be a brief summary of the results in the end of each section. It is hard to follow or get an idea of which trials failed completely and the reasons, which one advanced to the clinic or further (combination) trails (include the trail numbers), and what was the most promising treatment. For example, authors can add another column in the tables showing the next step, either proceed to clinic or ended without proceeding for each trial.

Thank you for your comments. According to the Clinical Trials gov, there were 85 clinical trials for ovarian clear cell carcinoma. Of these, 11 were for clear cell carcinoma only, 4 were complete, 1 was terminated, 4 were in recruitment, and 2 were pre-recruitment. On the other hand, there were more than 100 clinical trials that also included ovarian mucinous carcinoma, however GOG241 was the only clinical trial that included only mucinous carcinoma. None of the drugs used in these clinical trials outperformed standard treatment, and no drug has been approved for clinical use at this time. We have added summaries to Sections 2.2 and 3.3.

4,Authors should include rates of submission and success of clinical trials: how many percent of the drugs entering clinical trials that make it to clinical use? How many percent of the clinical trials for ovarian cancers is targeting clear cell and mucinous carcinomas?

Thank you for your comments. However, it is very difficult to survey all of the phase I-III clinical trials that have been conducted for ovarian clear cell carcinoma and mucinous carcinoma. We were not able to determine the correct submission and success rates of these clinical trials. The number of clinical trials mentioned above is only the result of what we were able to find out from Clinical Trials gov.

If there are any papers that describe these, please let me know so I can refer to them.

5,Authors mentioned the deficiency of primary diagnosis which can lead to the failure of treatments. They should propose new strategies for increasing accurate primary diagnosis in the conclusions.

Thank you for your comments. As shown in Table 5, from the viewpoint that ovarian mucinous carcinoma and colorectal carcinoma have similar biological characteristics, even if immuno-staining is added to HE-staining, it is difficult to deny colorectal cancer and accurately diagnose ovarian mucinous carcinoma. At this time, we believe that there is no clear evidence for new strategies to increase the accurate primary diagnosis of ovarian mucinous carcinoma.

Reviewer 3 Report

This well-written study assesses the available studies on mucinous and clear cell carcinomas of the ovary, which are rare neoplasms. The authors provide an overview on the current treatment options and molecular knowledge on the two histological types.

Comments are included below:

General comment:

  • The manuscript would benefit from shortening of descriptions of clinical studies and from expansion of the sections on biological background of both histological types. Were the whole exome or whole genome studies ever conducted? Or the majority of genomic studies use only a limited gene panel?
  • The names of the genes such as TP53, PIK3CA, have to be written in cursive

Section 2:

  • In the sub-section 2.1 on biological characteristics the authors should include the explanation on why clear cell carcinoma is suitable for immunotherapy. For example, high tumor mutation load of these tumors is probably the main reason.
  • Sub-section 2.2 should be named as “Previous clinical studies..”
  • Table 1:
  • Specify which types of mutations are frequent for each gene, for example missense, frameshift, and so on. The same goes for amplifications and methylation.
  • Lines 197-198: the reason behind better responsiveness for immunotherapy is that only 2 patients with clear cell carcinoma were included?
  • Line 200: “using PIK3CA mutations as biomarkers”. It is better to state using as a target
  • Line 206: Do the authors refer to cell lines?
  • Line 209: Omit “as they are promising drugs for clear cell carcinoma” treatment

Section 3:

  • Tables 4 and 5 are not relevant in the context of this study
  • Lines 291-293: the authors state that HER2 amplifications are relatively common which is in contrast with the only 18% of frequency reported in line 227
  • Lines 301-302: apart from the use of the molecular drugs, the studies on molecular landscape contribute enormously on identifying new therapeutic targets

Section 4:

  • This section has to summarize a little bit more the important facts from the previous sections, and somehow this has not been achieved completely.
  • Line 311: PI3K pathway ideally should be written in the same form throughout the entire manuscript, including the tables
  • Line 313-315: “Prognostic factors for ovarian mucinous carcinoma, including the KRAS and HER2-neu genes, MEK, and PI3K”. This phrase has to be modified as genes are not usually prognostic factors by themselves
  • Line 316: “targeting these prognostic markers”. It is better to state “targeting these genomic alterations”

Author Response

Response to Reviewer3

This well-written study assesses the available studies on mucinous and clear cell carcinomas of the ovary, which are rare neoplasms. The authors provide an overview on the current treatment options and molecular knowledge on the two histological types.

Comments are included below:

Thank you for reviewing our paper and for your appropriate comments. We responded to each comment. The revised parts of the text were also indicated in red.

  • The manuscript would benefit from shortening of descriptions of clinical studies and from expansion of the sections on biological background of both histological types. Were the whole exome or whole genome studies ever conducted? Or the majority of genomic studies use only a limited gene panel?

Thank you for your questions. Murakami et.al. (Ref18) and Itamochi et. al.. (Ref19) report the analysis of whole exome. However Amano et. al. (Ref20) used only a limited gene panel (http://www.broadinstitute.org/gsea/index.jsp).

  • The names of the genes such as TP53, PIK3CA, have to be written in cursive

Thank you for your advice. We have followed your advice and corrected there.

  • In the sub-section 2.1 on biological characteristics the authors should include the explanation on why clear cell carcinoma is suitable for immunotherapy. For example, high tumor mutation load of these tumors is probably the main reason.

Thank you for your comments. We added reasons why ovarian clear cell carcinomas are suitable for immunotherapy in the sub-section2.1.

  • Sub-section 2.2 should be named as “Previous clinical studies..”

Thank you for your advice. We have corrected it according to your suggestion.

  • Specify which types of mutations are frequent for each gene, for example missense, frameshift, and so on. The same goes for amplifications and methylation.

Thank you for your interest. Unfortunately, we were not able to find any literature that applies to your question. If you don't mind, I would like to ask you to refer me to the literature describing these.

  • Lines 197-198: the reason behind better responsiveness for immunotherapy is that only 2 patients with clear cell carcinoma were included?

Thank you for your question. Of the 20 patients enrolled in the study, 2 were clear cell carcinoma. one of the 2 patients had complete resolution, with a response rate of 50%. Therefore, it is effectiveness in analysis. Tumors with a mismatch repair deficiency (dMMR) phenotype respond well to immune checkpoint blockade therapy because they express many neoantigens with a high mutational load [21]. Therefore, ovarian clear cell carcinoma with ARID1A deficiency may benefit from immune checkpoint blockade therapy. We think this is another reason.

  • Line 200: “using PIK3CA mutations as biomarkers”. It is better to state using as a target

Thank you for your advice. We have corrected it according to your suggestion.

  • Line 206: Do the authors refer to cell lines?

Thank you for your question. This reference states that the cell lines were used.

  • Line 209: Omit “as they are promising drugs for clear cell carcinoma” treatment

Thank you for your pointing it out. We have removed it as you suggested.

  • Tables 4 and 5 are not relevant in the context of this study

Thank you for your comment. However we have created Tables 4 and 5 to understand the biological characteristics of ovarian mucinous and colorectal cancers. We believe that these tables will help the reader to understand better.

  • Lines 291-293: the authors state that HER2 amplifications are relatively common which is in contrast with the only 18% of frequency reported in line 227

Thank you for your advice. Your point is well taken. We have corrected the sentence.

  • Lines 301-302: apart from the use of the molecular drugs, the studies on molecular landscape contribute enormously on identifying new therapeutic targets

Thank you for pointing this out. We have learned a lot.

  • This section has to summarize a little bit more the important facts from the previous sections, and somehow this has not been achieved completely.

Thank you for your advice. Few clinical trials to date have shown better results than the standard of care. Therefore, we did not include them in Section 4, and focused on drugs that are expected to be new treatments in the future.

  • Line 311: PI3K pathway ideally should be written in the same form throughout the entire manuscript, including the tables

Thank you for your advice. We have corrected it `mTOR’.

  • Line 313-315: “Prognostic factors for ovarian mucinous carcinoma, including the KRAS and HER2-neu genes, MEK, and PI3K”. This phrase has to be modified as genes are not usually prognostic factors by themselves

Thank you for your advice. We have modified the prognostic factors to biomarkers, as you suggest.

  • Line 316: “targeting these prognostic markers”. It is better to state “targeting these genomic alterations”

Thank you for your advice. We have followed your advice and corrected it.

Round 2

Reviewer 1 Report

The authors have addressed my points with text revisions.

Reviewer 3 Report

The authors have addressed most of my suggestions with text revisions